behaviour

thanatology, death responses, non-human primates, chimpanzees

**Author for correspondence:**
Elizabeth V. Lonsdorf
e-mail: elizabeth.lonsdorf@fandm.edu

# Why chimpanzees carry dead infants: an empirical assessment of existing hypotheses

Elizabeth V. Lonsdorf[1,2], Michael L. Wilson[3],
Emily Boehm[4], Josephine Delaney-Soesman[2],
Tessa Grebey[2], Carson Murray[5], Kaitlin Wellens[5]
and Anne E. Pusey[4]

[1]Department of Psychology, and [2]Biological Foundations of Behavior Program, Franklin and Marshall College, Lancaster, PA, USA
[3]Departments of Anthropology and Ecology, Evolution and Behavior, University of Minnesota, Minneapolis, MN, USA
[4]Department of Evolutionary Anthropology, Duke University, Durham, NC, USA
[5]Center for the Advanced Study of Human Paleobiology, The George Washington University, Washington, DC, USA

EVL, 0000-0001-8057-401X; MLW, 0000-0003-3073-4518; CM, 0000-0003-2055-0061

The study of non-human primate thanatology has expanded dramatically in recent years as scientists seek to understand the evolutionary roots of human death concepts and practices. However, observations of how conspecifics respond to dead individuals are rare and highly variable. Mothers of several species of primate have been reported to carry and continue to interact with dead infants. Such interactions have been proposed to be related to maternal condition, attachment, environmental conditions or reflect a lack of awareness that the infant has died. Here, we tested these hypotheses using a dataset of cases of infant corpse carrying by chimpanzees in Gombe National Park, Tanzania ($n = 33$), the largest dataset of such cases in chimpanzees. We found that mothers carried infant corpses at high rates, despite behavioural evidence that they recognize that death has occurred. Median duration of carriage was 1.83 days (interquartile range = 1.03–3.59). Using an information theoretic approach, we found no support for any of the leading hypotheses for duration of continued carriage. We interpret these data in the context of recent discussions regarding what non-human primates understand about death.

# 1. Introduction

Philosopher Martin Heidegger [1, p. 107] argued that the capacity to understand death is uniquely human: 'Mortals are they who can experience death as death. Animals cannot do this.' In recent decades, detailed observations of animals in the wild and captivity have raised questions about the degree to which humans are unique in this capacity, warranting a broader understanding of the evolutionary roots of death concepts and practices [2,3]. Indeed, over the past several decades, the field of comparative thanatology has expanded dramatically and reports of non-human responses to death have begun to accumulate across taxa. For some species, these observations suggest an absence or a severely limited understanding of death. For example, in eusocial insects, chemical changes both prior to and after death act as death cues and initiate context-dependent responses, such as corpse removal, cannibalism, or burial [4]. While these responses in insects are flexible, they are thought be reflexive and to act primarily as disease or predator avoidance behaviours, rather than representing a cognitive understanding of death [3]. Similarly, crows (*Corvus brachyrhynchos*) occasionally contact dead conspecifics in exploratory, aggressive or sexual ways, which Swift & Marzluff [5] interpreted as an inability to recognize stimuli that indicate death, resulting in inappropriate or conflicting displacement activities.

Researchers have reported more complex responses to conspecific death in some mammals and some have argued that this reflects intensive maternal investment and/or greater cognitive capacities. For example, in a review of marine mammal cases, Reggente *et al.* [6] differentiate between non-cetaceans, which limit their behaviour to protecting the corpse from external attacks over a short period of time, and cetaceans, who show a longer period of post-mortem interactions and more nurturing behaviours, such as carrying and activities related to assisting with breathing. These differences in the degree of post-mortem interactions correspond to differences in degree of maternal investment. Within cetaceans, members of the Delphinidae family constitute the vast majority of cases of post-mortem interactions, with the highest incidence found among the *Sousa* and *Tursiops* genera [7]. These authors suggest that high encephalization may be an important predictor of the incidence of post-mortem interactions. Among land mammals, African elephants (*Loxodonta africana*) have been seen intensively investigating corpses from recently deceased individuals, attempting to lift or bury them, and repeatedly visiting a corpse for up to a week after death [8,9]. The preponderance of observations occurring in more socially complex taxa with extended maternal investment and large brains suggests that post-mortem death responses correlate with complex social cognition [10].

Non-human primates (hereafter: primates) are the best-studied taxa with regards to this topic (see [3] for a detailed review). Reported interactions have ranged from a relative lack of interest in a corpse in Verreaux's sifaka (*Propithecus verreauxi verreauxi* [11]) to behaviours interpreted as analogous to human stages of grief and understanding of death in captive chimpanzees (*Pan troglodytes* [12]). A fundamental issue when attempting to study wild primate responses to death is that ill or injured individuals often disappear before researchers can observe how others respond to deceased individuals. Thus, the available data are often reports of single or small numbers of events [13–15], which limits quantitative analysis and/or hypothesis testing. An exception is the continued carriage of and interaction with dead infants, which has been reported in many primates (reviewed in [16,17]), though only thoroughly described in a small number of species. For example, only one study has observed enough cases to estimate rates of carrying: Japanese macaque (*Macaca fuscata*) mothers carry 15% of all infants that die before 253 days of age (which was the oldest observed case) and 28.7% of infants that die within the first month of life [18]. In this study, we compiled the largest existing dataset of chimpanzee responses to infant corpses to determine carrying rates and examine four of the most commonly discussed hypotheses for continued interaction with deceased individuals.

While our understanding of primate mental states and cognitive abilities has advanced dramatically in recent years, our ability to empirically test how primates conceive of death is still extremely limited, especially in the wild (see [19], for a recent review). Of the several hypotheses circulating in the literature [16], there are four commonly discussed hypotheses regarding proximate explanations for the frequency and/or duration of infant corpse carrying (hereafter: ICC, table 1). The first two of these relate to the high degree of maternal investment in primates, but invoke different aspects of the mother–infant bond and result in opposing predictions. The *post-parturient condition hypothesis* was first proposed by Kaplan [20] and expanded on by Biro and colleagues [21], who presented three cases of extended carrying (all greater than or equal to 19 days) in chimpanzees in the context of mothers' post-parturient condition. They proposed that the maternal physiological conditions associated with pregnancy and birth facilitated continued care of corpses until lactation ceased and

**Table 1.** Existing hypotheses for the duration of infant corpse carrying (ICC).

| hypothesis (primary references) | measure | predicted relationship |
|---|---|---|
| post-parturient condition [20,21] | age of infant at death | ICC decreases with infant age at death |
| maternal-bond strength [16,18,22] | age of infant at death | ICC increases with infant age at death |
| slow decomposition [21,23] | season | ICC longer in the dry season |
| unawareness of death [24,25] maternal experience [17] | firstborn (Y/N) | ICC longer for firstborn offspring |
| unawareness of death [24,25] maternal age [17] | age of mother at infant death | ICC decreases with increased maternal experience |
| unawareness of death [24,25] context [17,26] | cause of death | external causes of death (e.g. infanticide) results in shorter ICC than internal causes (e.g. illness) |

cycling resumed, which then promoted abandonment of the corpse. This hypothesis predicts a higher likelihood or longer durations of ICC for newborn relative to older infants. However, a published case of ICC by a gelada mother (*Theropithecus gelada*) extended well past resumption of cycling and mating [23]. In addition, Li *et al.* [22] reported shorter periods of infant carrying for stillborn infants compared to an infant that died at one month of age in snub-nosed monkeys (*Rhinopithecus bieti*) and suggested that the length of bonding time prior to infant death may predict duration of ICC. Relatedly, Japanese macaque (*Macaca fuscata*) infants that died within a day of birth were less likely to be carried than those that lived several days [18]. These findings fall under the *maternal-bond strength hypothesis* [16], which predicts that older infants, who have had a longer (and presumably stronger) mother–infant bond, would be more likely to be carried and/or carried for longer than younger infants.

Researchers have also discussed continued interactions with dead infants as facilitated by environmental conditions, i.e. the *slow decomposition hypothesis*. Three of 14 cases of ICC in geladas extended beyond 10 days and resulted in mummification of the infant, prompting the authors to suggest that extended ICC may be facilitated by extremely dry climates that slow decomposition [23]. In support of this, the three longest cases of ICC in chimpanzees reported in [21] occurred during the dry season. However, extended instances of ICC have also been observed in multiple species during transitional and wet portions of the year, when decay proceeds more quickly [17,18,24,27,28].

Finally, the *unawareness of death hypothesis* [24,25] addresses whether mothers recognize that death has occurred. This hypothesis proposes that primate mothers might be unaware or unsure that their infant is actually dead, and that prolonged carriage represents a 'wait and see' period. Similarly, others have suggested that prolonged interaction with an infant corpse represents a period of information gathering about the state of the individual [29]. However, changes in maternal behaviour upon death, such as atypical carrying modes [17,18], cannibalism [30,31] or submergence of the corpse in water [28] suggest that mothers discriminate between alive and dead infants.

A recent meta-analysis [17] provided the first attempt to examine three of these hypotheses (post-parturient condition, slow decomposition and unawareness of death) simultaneously and across 18 different anthropoid primates. The authors used phylogenetic general linear mixed models to examine how specific factors predicted variation in ICC. According to the post-parturient condition hypothesis, ICC was predicted to decline with infant age, but infant age was not a significant predictor. Mean temperature did not predict ICC, but the authors did not examine humidity and rainfall, limiting evaluation of the slow decomposition hypothesis. To examine the unawareness of death hypotheses, they predicted that younger and/or primiparous mothers should carry dead infants longer as they have less experience with death and would therefore be more 'unsure'. However, they found the opposite; older mothers showed longer periods of carriage than younger mothers, but there were no differences between firstborn and later born offspring. Interestingly, the cause of death did predict the duration of ICC; infants that died of internal causes, such as sickness or stillbirth, were carried for longer than those that died of external causes, such as infanticide or electrocution. The authors speculated that these external causes may promote faster recognition of death due to the presence of

graphic injuries, and therefore faster cessation of carrying. Given the lack of strong support for any of the three hypotheses examined, and the oversimplification of life history and social variation necessitated by the phylogenetic approach, the authors recommended more detailed analyses of larger numbers of cases within a species when possible.

Rigorous examination of these hypotheses contributes to ongoing discussions regarding the uniqueness of human death concepts. Speece [32] proposed that a mature understanding of biological (versus spiritual) death in adult humans comprises four subcomponents: (i) irreversibility: death is permanent; (ii) non-functionality: dead individuals do not think, perceive or act; (iii) universality: all living things will die; and (iv) causality: death results from non-survivable bodily damage. Children appear to acquire most of these subcomponents by 7 years of age, although the 'causality' subcomponent is acquired later and acquisition is more variable [33–35]. Given that these concepts are necessarily expressed and studied by using language, attempts to determine which of these concepts (if any) are present in non-humans is challenging. In a recent review, Anderson [2] integrated findings from published reports of chimpanzee behaviour towards dead individuals with data regarding relevant cognitive abilities, including theory of mind, perspective-taking, inferential reasoning and self-awareness, to assess chimpanzees' understanding of death. He argued that mature chimpanzees, who have observed death and dying, probably understand that death is irreversible and that dead individuals are non-functional, but acknowledged that the evidence for universality and causality is more limited and/or has equally plausible alternative explanations.

In another recent review of non-human primate interactions with dead and dying individuals, Gonçalves & Carvalho [3] proposed a three-level model of death awareness that outlines the necessary cognitive abilities for a human-like understanding of death (see fig. 8 in [3]). This model proceeds from simple perceptual categorization at the lowest level to distinguish animate from inanimate, followed by the integration of sensory and contextual cues using associative mechanisms to discriminate between living and dead. At the most complex level, higher-order reasoning abilities contribute to a human-like death concept that includes the four components described above. They argued that the majority of evidence gathered thus far suggests that primates are capable of the first two levels, but that current evidence is more limited for the third level. In other words, there seems to be evidence for an understanding of 'dead', but less so for 'death' as a concept. However, they argue that the great apes, who exhibit some higher-order reasoning abilities, are among the animals most likely to possess emergent properties of a human-like death concept.

In this contribution, we examined the largest set of cases of ICC in chimpanzees, compiled from decades of research conducted at Gombe National Park, Tanzania. We used long-term data archives to identify and characterize the circumstances of death for all individuals known to have died or disappeared during infancy. First, we examined the rate of ICC to examine what factors predicted carriage versus no carriage. Next, we used an information theoretic approach to investigate which factors best explained variation in the duration of ICC in the context of the above-mentioned hypotheses regarding maternal condition, mother–infant bonding, environmental conditions and awareness of death (see table 1 for hypotheses and predictions). We also compiled the specific behavioural responses of mothers and others to infant corpses to evaluate the evidence for chimpanzees' understanding that death has occurred. Finally, we integrated our findings in the context of the recent reviews by Anderson [2] and Gonçalves & Carvalho [3].

# 2. Methods

## 2.1. Study site and long-term data collection

For this study, we analysed behavioural and demographic records from the long-term study of eastern chimpanzees (*Pan troglodytes schweinfurthii*) in Gombe National Park, a small (56 km$^2$) park located on the western border of Tanzania. Wild chimpanzees live in communities [36] that range in size from just a few individuals to over 200. Within these communities, subgrouping patterns are fission–fusion, wherein temporary subgroups form as a result of a combination of factors that may include food availability, sexual state of females and social relationships with other individuals [36,37]. Chimpanzee mothers are the primary carers for offspring, and close physical contact characterizes most of the first 2 years of life [38]. Offspring depend on their mother for nutrition through infancy until weaning between the ages of 3 and 5 years [39,40], but they continue to travel and socialize with their mother through the juvenile period, until at least the age of 8 years [41].

In this study, we focused on the two habituated communities of Kasekela and Mitumba, whose members are individually identified and matrilineal kin relationships are known for up to four generations. The Kasekela community was habituated in the early 1960s and focal follows on individual chimpanzees commenced in 1968, with standardized full-day follows beginning in 1973. Habituation of the Mitumba community began in the mid-1980s and full-day focal follows commenced in the mid-1990s [42]. In addition to the standardized behavioural data from focal follows, demographic records containing a variety of life-history variables are stored in a relational database created and maintained by Pusey [43]. From this database, we identified all recorded birth events that resulted in death prior to 5 years of age. To determine cause of death and responses to death, we screened focal behavioural data as well as multiple additional data sources for each individual (see electronic supplementary material, methods). We extracted information on the date the infant was last seen alive and any indicators of the cause of death. We assigned cause of death using categories adapted from two previous studies of mortality in this population [44,45]. In several cases, a mother simply appeared without her infant and death cause was categorized as 'disappeared' (see electronic supplementary material, tables S1 and S2).

## 2.2. Analyses of rate and duration of carriage

Following [18], we categorized each observed corpse as carried/not carried to examine the rate of ICC (i.e. the number carried out of the number of deaths). Due to the fission–fusion grouping dynamics, each individual chimpanzee cannot be observed every day, so complete data were not always available. Therefore, we took a conservative approach to estimate carrying duration and determined the minimum numbers of days that each infant was observed to be carried. To calculate the minimum carriage time, we extracted the date/time that the infant was first seen dead, the date/time of last observation of the corpse, and the date/time when the chimpanzee carrying the infant (most often the mother) discarded or was next seen without the corpse. If a mother was last seen carrying the corpse on a particular day, but not observed the following day and seen on next observation without the corpse, we estimated the endpoint as 06.00 on the morning following the last observation. We considered this appropriate given that multiple other carrying events were terminated early in the morning when a mother left the night nest without the corpse. In two cases, termination was noted as being 'during the day' without a specific time; we used 12.00 as the estimated endpoint for these cases. We converted total carriage time to days (or portions of days) for use in statistical analyses (following [17]).

To investigate which factors best explained variation in the duration of ICC, we used an information theoretic approach to compare candidate models containing one or a combination of our five main predictors (table 1) to a null model. We examined infant age at death (in years), to test the hypotheses related to post-parturient condition and maternal-bond strength. To test the slow decomposition hypothesis, we examined death season (wet = November through April, dry = May through October), on the assumption that decay occurs more quickly in the wet season. To examine the unawareness of death hypothesis, we tested variables associated with maternal experience: parity (whether the infant was firstborn or not) and maternal age at infant death (in years). In addition, we tested whether cause of death predicted duration of carriage. To reduce model complexity, we collapsed causes of death into five categories according to the 'attributed' causes in electronic supplementary material, table S1: inadequate maternal care; infanticide; never seen alive; poor health; and unknown.

We tested 11 models in total, each of which included a random effect for mother ID to account for multiple and unbalanced numbers of cases for each mother. The null model contained just mother ID. Five models contained each of the single parameters of interest. An additional model examined just contextual variables (season and cause of death). Four additional models examined a single context variable combined with a single mother or infant variable (table 3). Since our response variable (days carried) was continuous and skewed, we used generalized linear mixed-effects models with a gamma distribution and log link function, which provides better interpretability than using a log-transformed response variable [46]. We used only the subset of cases in which the birth mother was the primary carrier, given that (i) our aim was to compare hypotheses related to maternal condition against other sets of hypotheses, and (ii) it would be circular to analyse cases of carriers (i.e. non-mothers) whose behaviour (i.e. taking offspring) was the cause of death. We address non-mother carrying in further detail in the discussion. We removed one extreme outlier (APb1 who was carried for 15.70 days) that caused convergence problems for models containing parity, leaving us with a complete dataset for 22 cases. Additionally, the parity + season model failed to converge despite the use of an optimizer, and thus was removed from model comparison. Models were compared using the Akaike information

criterion corrected for small samples (AICc). We did not compare candidate models to a global model containing all predictors, given the likelihood of over-parametrization (i.e. examining five parameters simultaneously for 22 data points violates the recommended ratio of 10 data points per parameter [47]). We used base packages, the lme4 package [48] and the MuMIn package [49] in R v. 3.5.3 for all analyses.

## 2.3. Interactions with infant corpses

For cases in which sufficient data existed ($n = 28$), we extracted behavioural descriptions of community members' interactions with the corpse. The level of detail in these records was variable, given that some descriptions came from standardized data sources (e.g. focal follows) and some did not (e.g. ad lib observations). We focused specifically on a set of behaviours related to caregiving and investigation (see electronic supplementary material, methods for definitions).

# 3. Results

## 3.1. Rate of carrying and carry duration

We identified 93 cases of an infant being born into the population but not surviving to 5 years of age; an infant corpse was directly observed in 42 cases. However, in nine of these 42 cases, the mother was never observed in possession of the infant corpse. Seven of these nine cases represent instances of infanticide in which the mother never regained access to the corpse, and one represents a case of infanticide in which the attack was not observed, but the infant's body was later found. In this case, the mother may have carried the corpse for a period of time (if she regained access to it), but we have no way of knowing. The final case is that of the single individual that was killed by humans and whose body was recovered after the event [36]. The circumstances around this event are unknown, but it is plausible that the mother fled during the event and was not able to regain access to the corpse. In summary, in all observed cases in which a mother had access to her infant corpse, she carried it for a period of time.

We identified 33 cases of ICC, 30 of which had sufficient data to estimate the duration of carriage (table 2). The median duration of carriage was 1.83 days (interquartile range (IQR) = 1.03–3.59). Of these 33 instances, eight individuals were predominantly carried by an individual other than their mother. Three were orphans who were carried by an adopter who had shown caretaking behaviours to the orphan prior to death (two by older siblings and one by an unrelated adult female). Five of these were non-orphans who were taken by other adults, died, and were then carried by their abductor and others. When comparing the duration of cases for which we had sufficient data ($n = 7$ non-mothers compared to $n = 23$ mothers), we found no differences in duration of carriage (non-mothers: median = 1.83 day, IQR = 1.48–2.72; mothers: median = 1.75, IQR = 0.923–4.57); Wilcoxon rank-sum test: $W = 83$, $p = 0.922$).

## 3.2. Predictors of carry duration

A summary of the model comparison results with model-averaged parameter estimates and confidence intervals is shown in table 3. None of the models performed significantly better than the null model according to a ΔAICc criterion of 2 and all model-averaged parameter estimates included 0 in the 95% confidence interval.

## 3.3. Interactions with infant corpses

We extracted behavioural descriptions of interactions with infant corpses and compiled these separately for mothers/adoptive mothers, siblings and others (table 4). Given that the level of detail in these records was variable and collected using different methods (see electronic supplementary material, methods), we were certain about the presence of a behaviour that was reported, but not about absences of behaviours. As such, we restricted our analyses to qualitative descriptions of behavioural patterns. A clear pattern was that nearly all mothers/adoptive mothers exhibited atypical carrying postures, and this often happened within a few hours of death. Typical infant transport comprises either riding ventrally (for very young infants) or dorsally (for infants greater than 1 year of age) [38,50]. However, infant corpses were transported in the mouth, a hand, the groin pocket (between the hindlimbs and torso), the neck pocket (between the jaw and shoulder), slung over the shoulder or around the neck (figure 1), or dragged on the ground behind. These modes of carrying were also different from those

**Table 2.** All cases of infant corpse carrying by ID with associated variables. Only those with included = Y used in statistical analyses.

| ID | mother ID | carrier ID | sex | date of observation | infant age at death (years) | mother age (years) | firstborn | season | cause of death | carry duration (days) | included |
|---|---|---|---|---|---|---|---|---|---|---|---|
| APb1 | AP | AP | U | 27 Aug 90 | 0.238 | 17.18 | Y | dry | unknown | 15.7 | N |
| BH | MB | MB | F | 18 Sep 71 | 0.260 | 26.21 | N | dry | poor health | 4.79 | Y |
| BIMB1 | BIM | BIM | U | 1 May 07 | 0.008 | 14.87 | Y | dry | poor health | 0.615 | Y |
| DYL | DL | DL | M | 12 Nov 04 | 1.610 | 18.41 | Y | wet | unknown | 1.89 | Y |
| FI | FF | FI | M | 29 July 97 | 0.895 | 39.07 | N | dry | poor health | 0.371 | Y |
| FIT | FLI | FLI | F | 02 Oct 16 | 1.385 | 18.20 | Y | dry | unknown | unk | N |
| FNB1 | FN | FO | M | 09 Oct 14 | 0.115 | 33.55 | N | dry | inadequate maternal care | 1.14 | N |
| GAB2 | GA | GA | M | 23 July 08 | 0.014 | 15.44 | N | dry | inadequate maternal care | 1.17 | Y |
| GAB3 | GA | GA | M | 31 July 08 | 0.036 | 15.46 | N | dry | inadequate maternal care | 3.87 | Y |
| GASB | GA | GA | M | 05 May 07 | 0.000 | 14.22 | N | dry | never seen alive | 0.609 | Y |
| GLIB1 | GLI | GM | F | 21 Jan 12 | 0.534 | 13.53 | Y | wet | inadequate maternal care | 2.13 | N |
| GMB1 | GM | GM | M | 22 May 87 | 0.019 | 16.51 | N | dry | poor health | 1.25 | Y |
| GO | ML | ML | M | 11 Oct 86 | 1.284 | 37.28 | N | dry | poor health | 0.762 | Y |
| GOD | GA | GM | M | 25 Sep 06 | 0.411 | 13.61 | Y | dry | inadequate maternal care | 1.82 | N |
| GR | OL | OL | M | 13 Sep 66 | 0.101 | 29.20 | N | dry | poor health | 1.74 | Y |
| GY | ML | ML | M | 12 Aug 78 | 0.808 | 29.11 | N | dry | poor health | 1.94 | Y |
| JE | MD | MD | F | 14 Feb 65 | 0.235 | 15.62 | Y | wet | infanticide | unk | N |
| JNS1 | JNM | JN | U | 30 June 75 | 1.492 | n.a. | U | dry | orphan | unk | N |
| KN | KD | KD | M | 10 Jan 91 | 1.489 | 24.61 | N | wet | poor health | 4.57 | Y |
| MKWB1 | MKW | VAN | M | 09 Oct 14 | 0.082 | 20.27 | U | dry | inadequate maternal care | 11.5 | N |
| MLB1 | ML | ML | F | 18 Nov 69 | 0.000 | 20.38 | N | wet | never seen alive | 2.76 | Y |
| MLB2 | ML | ML | M | 08 Jan 76 | 0.005 | 26.52 | N | wet | infanticide | 1.75 | Y |

(*Continued.*)

**Table 2.** (*Continued.*)

| ID | mother ID | carrier ID | sex | date of observation | infant age at death (years) | mother age (years) | firstborn | season | cause of death | carry duration (days) | included |
|---|---|---|---|---|---|---|---|---|---|---|---|
| OLB1 | OL | OL | U | 07 Sep 67 | 0.000 | 30.18 | N | dry | never seen alive | 0.910 | Y |
| PIB1 | PI | PI | M | 21 Apr 78 | 0.016 | 16.80 | Y | wet | inadequate maternal care | 0.923 | Y |
| PIB2 | PI | PI | U | 08 July 88 | 0.000 | 27.02 | N | dry | never seen alive | 0.986 | Y |
| PT | PL | PL | M | 17 Apr 73 | 2.609 | 20.79 | Y | wet | poor health | 2.13 | Y |
| SHO | RAF | KON | F | 19 Apr 96 | 1.068 | n.a. | N | wet | orphan | 1.83 | N |
| SR1 | S | S | F | 12 Feb 68 | 1.224 | n.a. | N | wet | orphan | 0.0208 | N |
| SV | SW | SW | M | 07 Mar 90 | 0.832 | 31.68 | N | wet | poor health | 8.75 | Y |
| TAR | TG | TG | F | 24 Aug 13 | 0.879 | 24.34 | N | dry | infanticide | 1.18 | Y |
| TUK | TG | TG | M | 31 July 15 | 0.808 | 26.27 | N | dry | infanticide | 8.23 | Y |
| TZB2 | TZ | SA | M | 12 Mar 15 | 0.537 | 36.69 | N | wet | inadequate maternal care | 2.71 | N |
| VANB1 | VAN | VAN | U | 27 July 16 | 0.233 | 28.07 | N | wet | never seen alive | 6.65 | Y |

**Table 3.** Summary of model comparison for duration of dead infant carrying. Parameters include the intercept ($\beta$); model estimates for infant age at death (in years); whether the infant was firstborn; mother age at death; death season; and cause of death (collapsed categories; table 4); the Akaike information criterion (corrected for small samples: AICc); the difference in AICc between the model and the best model ($\Delta$); and model weight (w). Models are arranged in order from best (lowest AICc) to worst (highest AICc). Model-averaged parameter estimates (MAP) with upper (97.5%) and lower (2.5%) bounds of the 95% confidence intervals are given in the bottom rows. *Estimates for individual COD categories provided in electronic supplementary material, table S3.

| corresponding hypothesis | $\beta$ | infant age (yrs) | firstborn (Y) | mother age (yrs) | season (wet) | cause of death (COD) | AICc | $\Delta$ | w |
|---|---|---|---|---|---|---|---|---|---|
| null | 0.712 | | | | | | 89.3 | 0.00 | 0.358 |
| slow decomposition | 0.477 | | | | 0.589 | | 89.7 | 0.391 | 0.295 |
| unawareness (maternal experience) | 0.848 | | −0.480 | | | | 91.5 | 2.20 | 0.119 |
| unawareness (maternal age) | 1.12 | | | −0.0172 | | | 92.0 | 2.72 | 0.092 |
| post-parturient condition/maternal-bond strength | 0.757 | −0.0759 | | | | | 92.2 | 2.97 | 0.081 |
| post-parturient condition/maternal-bond strength + slow decomposition | 0.499 | −0.0356 | | | 0.590 | | 93.0 | 3.77 | 0.054 |
| unawareness (COD) | 1.27 | | | | | * | 103 | 13.3 | 0.00 |
| unawareness (COD) + unawareness (maternal experience) | 1.823 | | −0.9913 | | | * | 105 | 15.9 | 0.00 |
| unawareness (COD) + slow decomposition | 0.689 | | | | 0.641 | * | 106 | 16.4 | 0.00 |
| unawareness (COD) + post-parturient condition/maternal-bond strength | 1.280 | −0.0245 | | | | * | 108 | 18.4 | 0.00 |
| MAP | 0.689 | −0.00809 | −0.0574 | −0.00158 | 0.206 | * | | | |
| 2.5% | −0.170 | −0.262 | −0.544 | −0.0234 | −0.502 | * | | | |
| 97.5% | 1.55 | 0.245 | 0.429 | 0.0202 | 0.913 | * | | | |

**Table 4.** Observed interactions of mothers and others with infant corpses. M, mothers, A, adoptive mothers, S, sibling, O, others. Numbers in () represent numbers of individuals if more than one. 'nd' for atypical transport means that the infant was carried but specific mode of transport was not recorded.

| ID | Mothers/Adopter | | | | | | | | | | | Other interactants | | | | | | | | | | | | | Total interactants |
|---|---|---|---|---|---|---|---|---|---|---|---|---|---|---|---|---|---|---|---|---|---|---|---|---|---|
| | Atypical transport | Peer | Smell | Inspect body | Inspect face | Inspect genitals/anus | Manip-ulate | Groom | Rough handling | Swat flies | Eat | Atypical transport | Peer | Smell | Inspect body | Inspect face | Inspect genitals/anus | Manip-ulate | Groom | Sexual | Play | Rough handling | Chase flies | Eat | |
| APb1 | M | | | | | | | | | | | | | | | | | | | | | | | | 1 |
| BH | nd | | M | | | | | | | | | S | O | | S | | | S | S | | | | S | | 3 |
| BIMB1 | M | | | | | | | | | | | | | | | | | | | | | | | | 1 |
| DYL | M | | M | | | | | M | | | | | O (4) | O (3) | | | | O (x2) | O | | | | | | 5 |
| FI | M | | M | | | | | | | M | | S | S (2), O (4) | S (3), O (2) | S (3) | S | | S | S | | S | | | S | 8 |
| GAB2 | A; M | A; M | | M | | | | | | A | | | O (2) | | | | | | | | | | | | 4 |
| GAB3 | A; M | M | | | M | | | M | | M | | | O | | O | | | O | | | | O | | | 3 |
| GASB | nd | M | | | | | | | | | | O (5) | O (5) | O (3) | | | | O (2) | | | O (3) | O (2) | | O (2) | 7 |
| GLIB1 | A | | | | | | | | M | | | O | | | | | | | | | O (2) | O | | | 3 |
| GMB1 | M | | | | | M | M | M | M | M | | | | | | | S | | S | | | | | | 2 |
| GO | M | | | M | | | | | | M | | | | O | O | | | | | | | S | | | 2 |
| GR | M | | M | M | | M | | | | M | | S | S, O (5) | S | S | S | S, O | | S | | | S | S | | 7 |
| GY | M | | M | | M | | | M | | M | | S | O (7) | O (2) | S, O | S | S, O | S | S, O | | | S | S, O (2) | | 9 |
| JE | M | | M | | | | | | | M | | S | O | S | S | | | | S | | | | | | 2 |
| KN | M | M | M | M | M | | | M | | M | | S | S, O | O | S | | | | S | | | | S, O (2) | | 4 |
| MKWB1 | A | | | | | | | | | | | | | O | O | | | | | | | | | | 2 |
| MLB1 | M | M | M | M | M | M | M | M | | M | | | S, O (2) | | | | | | | | | | S | | 4 |
| MLB2 | M | M | M | | | | | M | | | | | S, O | | | | | O | S | | | | | | 3 |
| OLB1 | M | | | M | M | | | | | | | O (2) | O (5) | O (2) | O (2) | | | O | | O | O (2) | | | | 6 |
| PIB1 | M | M | M | M | M | | M | M | | M | | | | | | | | | | | | | | | 1 |
| PIB2 | M | | | M | | | M | M | | M | | | O, S | | | | | | | | | | | | 3 |
| PT | M | | | | | | | M | | | | | | | | | | | | | | | | | 1 |
| SR1 | A | | A | | | | | | | | | | | O | | | | | | | | | | | 2 |
| SV | M | | | | | | | | | | | | | O (4) | | | | | | | | | | | 5 |
| TAR | nd | M | | | | | | | | | M | | S, O | | | | | | | | | O | | S (2), O | 5 |
| TUK | M | | | | | | | | | | M | S | O | | | | | | | | | | | S | 3 |
| TZB2 | A | | | | | | | A | | | | | | | | | | O | | | | | | | 2 |
| VANB1 | M | | | | | | | M | | | | | | | | | | | | | | | | | 1 |

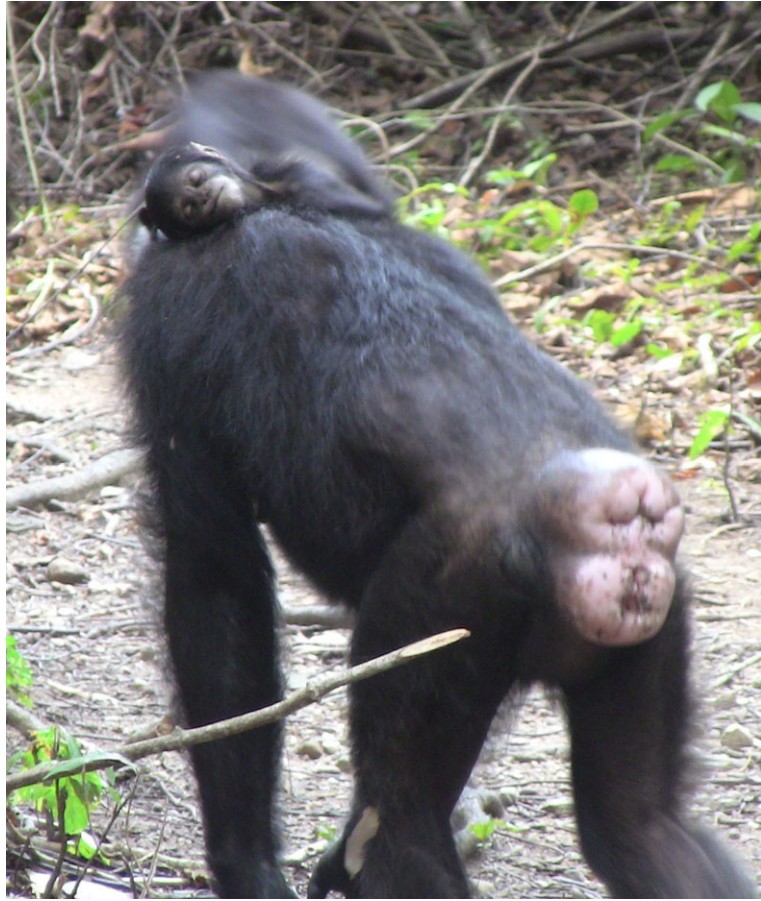

**Figure 1.** Example of an atypical carrying posture (draped around the neck) used to transport an infant corpse.

used to transport an ill and/or injured infant, who are most often held ventrally and supported with a hand while the mother walks tripedally [36].

Another consistent pattern is that siblings were the most likely to interact with a corpse after the mother. Both mothers and siblings groomed the corpse post-mortem more often than other individuals did. Investigatory behaviours with contact (inspect face, body, genitals/anus and manipulate) were also performed largely by mothers and siblings, but other individuals did this as well. Mothers were never observed directing play or exhibiting sexual behaviours towards the corpse, while siblings and others did. Rough handling was reported more rarely overall and there was only one instance reported of rough handing by a mother: GM was seen to throw the corpse of GMb1 on the ground directly in front of other chimpanzees three separate times. A single individual (TG) was the only mother seen to eat part of her own infant. She did so for two different infants and in both cases, this was after the infant was killed by other chimpanzees. In the first case, the infant TAR was killed immediately and the mother, two siblings and the attacker began feeding on the corpse soon after death. In the second case, the infant TUK was mortally wounded during an attack but was recovered by TG before he subsequently died. TG carried the corpse for a minimum of 5 days before she and her other offspring were seen feeding on it. Instances of other individuals eating portions of a corpse outside of the infanticide context happened in two cases. A stillborn infant corpse (GAsb) was taken from the mother and partially eaten by others in the community before the mother retrieved the corpse. Infant FI died during an epidemic of sarcoptic mange, and his brother was seen to bite off and eat a piece of his corpse. Non-contact investigatory behaviours such as peering at and smelling the corpse were the most common behaviours performed by non-mothers. See electronic supplementary material, results for example narrative descriptions.

## 4. Discussion

In this contribution, we presented the largest systematic and quantitative study of variation in infant corpse carrying in wild chimpanzees. We found that chimpanzees carry infant corpses at high rates,

despite behavioural evidence that they recognize that death has occurred. Moreover, none of the variables we examined to test existing hypotheses for the duration of ICC were significant.

Recent discussions surrounding the phenomenon of infant corpse carrying have recommended reporting the rate of corpse carrying [18] and what factors contribute to variability in rates [16]. We found that in every recorded case in which a mother was able to access her infant's body, she carried it. The only exceptions to this pattern were when the infant was killed by infanticide and the mother never regained access, and when an infant corpse was recovered in the field and the mother was not seen (with or without the corpse) in the interim. It is possible that the disappearances represented cases in which infants were not carried, but given that we cannot see every chimpanzee every day, it is also possible that they were all carried. Even if we assume that all of the 41 individuals that disappeared were not carried, and include the two cases in which corpses were recovered without ever being seen with their mother, the carrying rate would be 50/93 or 54%. What is clear is that rate of carrying we report here, between 54% and 100%, is substantially higher than that reported in Japanese macaques in which 15% of all corpses are carried and 28.7% of very young infants (less than or equal to 30 days old) are carried [18]. However, complete information regarding what happened to macaque infants that were not carried (i.e. whether corpses were observed but abandoned immediately or whether the infant simply disappeared/was subject to predation, etc.) was not reported. Such information would allow us to more fully understand the differences in carrying rates we report here. Comparable rate data are not yet available for other primate populations, and should be a focus of ongoing data collection.

We documented 33 cases of dead infant carrying, ranging in length from 30 min to over 15 days. On average, infant corpses were carried for a little over 3 days, and only two were seen to be carried over 10 days. Both of these carriers resumed cycling and mating while still carrying the corpse. In 24% of instances, the primary carrier was a non-mother but there was no difference in the duration of carriage between mothers and non-mothers. Interestingly, one of the relatively prolonged cases was performed by a non-mother who abducted a live infant and then carried it after it died. The mean carrying duration we found for Gombe was shorter than previously reported for wild chimpanzees. A recent meta-analysis of cases in anthropoid primates [17] reported a mean duration for chimpanzees of 32.94 days, as well as the largest within-species variance among the other species examined, from 2 to 114 days. These authors included only chimpanzee cases extracted from previously published reports ($n = 9$), so it is possible that the published literature is biased towards extremely long cases. However, a compilation of 14 cases from Mahale [27] also reports a larger range of durations, from 30 min to approximately 126 days. An interesting possibility is that there may be site-specific cultural differences in responses to death [2,21] as there are with several other behaviours [51]. Alternatively, differences in ranging patterns may affect duration of carriage such that larger day ranges constrain duration of carriage, as suggested by Carter *et al.* [52] for chacma baboons (*Papio ursinus*).

Several proximate explanations have been put forth to explain either the likelihood or duration of ICC. We examined infant age at death to test the post-parturient condition hypothesis and the maternal-bond strength hypothesis, but none of our candidate models that contained infant age at death provided a better fit to our data than the null model (table 3). Moreover, we documented several instances of carrying by non-mothers as well as two cases of prolonged carrying in which the carriers resumed cycling and mating while still carrying the corpse (as also seen in geladas [23]). The only notable age-related pattern we found was that no infants over 3 years of age were observed being carried. However, there were only nine individuals between the ages of 3 and 5 years of age in the full dataset, five of which disappeared (see electronic supplementary material, table S2). Of the remaining four, one was orphaned, one was killed by chimpanzees, one was killed by humans, and one died of illness. Thus, only the infant that died of illness could have been carried, but there were 6 days between sightings of this mother with and without the infant, so carrying cannot be determined. One possibility is that at 3 years of age, which is when most offspring cease riding on their mothers [50], corpses are simply too heavy and/or awkward to be carried for lengthy periods of time. A similar age effect was reported in Japanese macaques, in which carrying rates increased with age up until 30 days, but then decreased [18]. In sum, the small number of deaths above age three and the uncertainty around them prevent determination of whether there is an upper age limit for an infant to be carried, but the combined evidence does not support the post-parturient condition hypothesis of infant carrying. Bond strength probably plays a role in *who* carries a corpse given that mothers, adoptive mothers and siblings exhibited the most carrying (table 4), but it does not appear to predict duration of carriage in a consistent way.

The slow decomposition hypothesis [23] posits that prolonged infant carrying is facilitated by dry climates and/or in the dry parts of the year. According to our model comparison, the model including season was the second-highest-weighted model (table 3), although this model performed no better than

the null model. In fact, while our two longest cases (table 2) occurred during the dry season, carrying duration tended to be longer in the wet season (though this parameter estimate included 0 in the 95% CI). Cases of prolonged carrying at Bossou occurred during the dry season [21], but prolonged carries in Mahale happened throughout the year [27]. In sum, we found no clear support for the hypothesis that carrying duration varied according to within-site environmental conditions.

The unawareness of death hypothesis concerns what mothers may or may not understand about death [24,25]. One prediction arising from this hypothesis is that younger and/or first-time mothers carry dead infants for longer in order to gather more information and be certain the infant is dead. None of our candidate models that included either firstborn status or mother's age at death performed significantly better than the null model. In fact, aside from one outlier (APb1), mothers tended to carry firstborn offspring for shorter durations than later born offspring (though this parameter estimate included 0 in the 95% CI). Given that we only have nine firstborn offspring in the dataset of observed corpses, more cases are needed to confirm this pattern. In the above-mentioned anthropoid primate meta-analysis [17], younger females exhibited shorter durations of carrying (regardless of firstborn status) in contrast to prediction. However, because datasets from across 18 different species were combined, mother's age was partitioned into two categories, younger versus older (according to median life span for the species). That difference in methodology, coupled with the smaller sample of nine chimpanzees in that study, may account for our different findings. These authors also found that infants that died of sickness or were stillborn were carried for longer than those that died of infanticide or unnatural causes (such as electrocution) and suggested that more violent/obvious causes of death resulted in faster detection of death and abandonment of corpses. By contrast, we found that candidate models including cause of death provided no better fit than the null model.

Our examination of the specific types of behaviour directed at the corpse suggests that mothers/primary carriers rapidly recognized a change of state in the infant, given that atypical carrying postures were exhibited soon after death. In fact, these postures more closely resembled those used for objects than live infants [53,54] and would be uncomfortable or harmful for a live infant, suggesting that mothers persisted in infant corpse carrying for hours or days after recognizing a change in state. There are also multiple sensory death cues: failure to respond to tactile stimulation, olfactory cues of putrefaction, and the presence of flies. While we do not have consistent quantitative measures of distance for our cases, observers often described a process in which the mother gradually increased her distance from the corpse, as was reported in [29]. As the post-mortem period progressed, siblings and others were permitted to interact with the corpse in a manner similar to object play, or roughly handle the corpse. In very rare cases, mothers exhibited rough handling or ate parts of the corpse themselves. These qualitative reports and our model comparison results do not support the unawareness of death hypothesis.

With regards to the broader question of whether there is evidence of a human-like death concept in chimpanzees, our data provide additional evidence that chimpanzees understand at least two of the four subcomponents reviewed in [2]: non-functionality, given the changes in behaviour described above, and irreversibility, given the eventual abandonment of the corpse. Whether or not chimpanzees understand death as universal is not possible to examine with our data, and indeed, is difficult to explore without a common language. The causality component is the last to be acquired by human children and its importance is debated given its reliance on knowledge of biological processes that have only come about through modern science and medicine (A Gonçalves 2019, personal communication). Das *et al.* [17] argued that anthropoid primates comprehend causality, because infants that died of unnatural and externally observable causes were carried for shorter periods than those that died of natural causes. However, we did not find that cause of death explained the variation in duration of infant carrying in our larger, single-species sample.

Evaluating our findings according to the three-level model of death awareness proposed by Gonçalves & Carvahlo [3], our data provide support for the first two and suggest that chimpanzees quickly distinguish between animate and inanimate and also integrate sensory and contextual cues to discriminate between living and dead. The third level of death awareness encompasses the four subcomponents of a human-like death concept described above, and we have provided additional evidence in accordance with [2] for chimpanzees' understanding of non-functionality and irreversibility. Evidence for the subcomponents of universality and causality will probably need to come from rigorous and ethical cognitive experiments conducted in captive settings [19].

Given that we found no evidence for the main existing hypotheses for ICC, the question remains as to why chimpanzee mothers, if they have distinguished between alive and dead, continue to carry their infant corpses for multiple days after death and often exhibit caretaking behaviours such as grooming. This behaviour could be interpreted as lack of awareness that death has occurred, but the behavioural changes we described above suggest otherwise. An intriguing possibility is that infant corpse carrying represents a primate analogue of human grief [17,55]. However, grief has been difficult to systematically

define and operationalize; even in humans, grief is recognized as a 'highly individualized and dynamic process' [56, p. 119]. Evidence in non-verbal animals must come from behaviour, and an increasing number of species have been reported to exhibit behaviours similar to humans. These include prolonged association with the corpse [8,57–59], decreased appetite [12,17,57,60], and social isolation and/or avoidance [17,22,57]. How non-human animal emotions are defined, measured and whether they differ in degree or kind from human emotions is an active area of scientific debate (see [61,62]). Responses to death may be a particularly fruitful area of focus for the study of primate emotions.

Much work remains to be done to achieve a more complete understanding of non-human animal responses to death. We echo the calls of our colleagues [3,16] for more detailed descriptions of cases using standardized terminology and variables, and collection of physiological data such as stress metabolites. Detailed analyses of changes in maternal activities budgets, such as reduced feeding and socializing, as well as others' consolation behaviours towards mothers, will help to provide a more complete picture of a species' understanding of death. These data are extraordinarily difficult to collect and accumulate given the rarity of observed deaths in the wild, which highlights the importance of long-term studies for understanding of the complexity of animal behaviour. While many questions remain, observations reported here and elsewhere challenge Heidegger's [1] assertion that 'Animals cannot do this'.

Ethics. This work was observational and non-invasive, and complies with the Association for the Study of Animal Behaviour Guidelines for the Use of Animals in Research. In addition, all work complied with the legal and permitting requirements of the government of Tanzania, where the work was carried out.

Data accessibility. The datasets supporting this article are available in the electronic supplementary material.

Authors' contribution. E.V.L. conceived and designed the study, analysed the data and drafted the manuscript. M.L.W. and A.E.P. helped to design the study and contributed data. E.B., J.D.-S., T.G., C.M. and K.W. contributed data and assisted with data acquisition. All authors were involved in revising the manuscript and approving of its final form.

Competing interests. The authors declare no competing interests.

Funding. This work was supported by funds from the National Institutes of Health (grant no. 5R00HD057992, R01 AI50529, R01 AI58715), the National Science Foundation (grant nos. DBS-9021946, SBR-9319909, BCS-0452315, IOS LTREB-1052693, 1457260 BCS-1753437), the Leo S. Guthman Foundation, the National Geographic Society, the Harris Steel Group, the Windibrow Foundation, the Carnegie Corporation, the University of Minnesota, Franklin and Marshall College, George Washington University and Duke University. Funding for open access provided by the Franklin & Marshall College Open Access Publishing Fund.

Acknowledgements. The authors thank Jane Goodall, the Jane Goodall Institute and Tanzania National Parks (TANAPA) for initiating and continuing the over 59-year research tradition at Gombe. We are extremely grateful to the Gombe Stream Research Centre assistants for data collection. Thanks are also due to Anthony Collins and Deus Mjungu for their continued dedication to long-term data collection at Gombe and to Shannon Rovias for assistance with case extraction from the data archive. Special thanks to Hank Klein, Maria Botero, Emily Wroblewski and Kelly Ostrofsky for sharing their field notes for key cases and to Margaret Stanton for statistical advice. Permission and support to carry out research at Gombe was granted by the Government of Tanzania, Tanzania National Parks, Tanzania Commission for Science and Technology and Tanzania Wildlife Research Institute. Special thanks to the Center for Advanced Study in the Behavioral Sciences and the fellowship class of 2018–2019 for support and conversations that greatly improved this work.

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
