## [Reviewer comments · Royal Society Open Science]

Review History

Decision letter (RSOS-200931.R0)

Dear Dr Lonsdorf:

It is a pleasure to accept your manuscript entitled "Why chimpanzees carry dead infants: an empirical assessment of existing hypotheses" in its current form for publication in Royal Society Open Science. The comments of the reviewer(s) who reviewed your manuscript are included at the foot of this letter.

You can expect to receive a proof of your article in the near future. Please contact the editorial office (openscience_proofs@royalsociety.org) and the production office

(openscience@royalsociety.org) to let us know if you are likely to be away from e-mail contact -- if you are going to be away, please nominate a co-author (if available) to manage the proofing process, and ensure they are copied into your email to the journal.

on behalf of Dr Alecia Carter (Associate Editor) and Professor Kevin Padian (Subject Editor).

Associate Editor Dr Alecia Carter Comments to Author:

Associate Editor
Comments to the Author:
Decision on RSOS-200931

Dear authors,

I hope this finds you all well in these disturbing and difficult times.

I was excited to receive and read your manuscript not least because it takes a timely and rigorous hypothesis-testing approach to ICC. Overall, I found your manuscript to be well-written and well-executed. I have a few minor comments and some typos to correct, but otherwise I feel that, as the reviewer's comments have been thoroughly dealt with, this manuscript does not require further review and I look forward to seeing it published.

Minor comments:

L33: carried carry -> carried

LL33-34: I'm afraid I disagree with the referee that this result should be highlighted in the abstract like this. It is misleading. Yes, the carrying rates are higher, but as written, this implies that chimps 'choose' to carry more often than e.g. monkeys (well, one monkey, given the data available in this field). This may be true, but, crucially, we do not have the data to support this. The fact is that in most cases, observers do not witness the moment of an infant's death and so we have no idea whether the corpse was 'available' to be carried or not, and whether the mother chooses to abandon the corpse on its death or not. It is entirely possible that predation is much higher for smaller-bodied primates like macaques, and carrying rates much lower than in chimpanzees because there is no corpse to carry. Or macaques may carry for shorter durations, which are missed by observers (30 min is a very short period of time, and could easily be missed). While this nuance is included in the discussion, it is missed in the abstract and I would argue that it implies something the authors' data cannot support. Please change it here.

LL35, 284: The data are not normally distributed and are bound by 0, so the median and IQR should be presented (as has been done at LL291-292). As it is, this suggests that >1/4 of cases (i.e. cases below -1 S.D.) are carried for less than 0 days, which is not possible.

LL58-61: This is a bit confusing. Is Reggente et al. comparing mammalian responses? Is so, what about primates' and proboscids' responses? As it's written, it seems as if Reggente et al. assert that nurturant behaviour towards dead conspecifics is limited to cetaceans, but this is not the case (in their paper, or in the literature). Please consider re-phrasing.

para starting L299: this paragraph is far too long. Please consider breaking it up.

L310: that -> than

LL349-353: Please be careful with this assertion. (See comment LL33-34 above.) It is possible that all Japanese macaque mothers would have carried their infants if they had access to the corpses (i.e. the carry rate could be between 15% and 100%), and this should be acknowledged. Indeed, carriage may be lower for infants >30 d old because predation is higher for infants who are not being carried by their mothers. (For the record, I suspect that it is true that chimps carry at higher rates, but we just cannot say this with certainty. Please provide all the caveats.)

L364: hyphenate "previously published"

LL401-402: My understanding of the climate hypothesis is that it was proposed to explain between-population variation in corpse carriage. While I think it's fine to test it at the within-population level, this should be recognised in this concluding sentence by adding "within-site" between "according to" and "environmental conditions".

L403: "chimpanzees" is very specific, particularly since the references here are broader than just chimpanzees. Perhaps change to "primates" or "mammals" or "animals"

In the referee responses:

Q23: Heightened cortisol is a typical human response to grief, so I feel this is an appropriate link to make. (Indeed, it would be the first physiological indicator that I would search for when looking for evidence of a homologous response to bereavement in non-human primates.) Since the authors disagree, I'm not asking for this study to be cited, nor am I asking for any change to be made. I just want to highlight that, in this case, I agree with the reviewer.
